# Validation of the Concise Assessment Scale for Children’s Handwriting (BHK) in an Italian Population

**DOI:** 10.3390/children10020223

**Published:** 2023-01-27

**Authors:** Annalivia Loizzo, Valerio Zaccaria, Barbara Caravale, Carlo Di Brina

**Affiliations:** 1Centro di Riabilitazione Didasco, Via Pescosolido 160, 00158 Rome, Italy; 2Department of Human Neurosciences, Sapienza University of Rome, Via dei Sabelli 108, 00185 Rome, Italy; 3Department of Developmental and Social Psychology, Sapienza University of Rome, Via dei Marsi 78, 00185 Rome, Italy

**Keywords:** BHK, handwriting skills, handwriting difficulties, primary school children

## Abstract

Handwriting difficulties represent a common complaint among children and may cause a significant delay in motor skills achievement. The Concise Assessment Scale for Children’s Handwriting (BHK) assesses handwriting skill in clinical and experimental settings, providing a quick evaluation of handwriting quality and speed through a copying text. The aim of the present study was to validate the Italian adaptation of the BHK in a representative primary school population. Overall, 562 children aged 7–11 from 16 public primary schools of Rome were included and asked to copy a text in 5 min using cursive handwriting. Handwriting quality and copying speed were measured. The included population followed a normal distribution for the BHK quality scores. Sex influenced the total quality scores, whereas school level influenced the copying speed. The BHK quality score was higher in girls (*p* < 0.05) and resulted as a stable parameter along the school years, without significant variations with regard to the years spent in handwriting exercise (*p* = 0.76). The handwriting speed was influenced by school level, and significant differences were found for each of the grades from the second to the fifth (*p* < 0.05), but not for gender (*p* = 0.47). Both BHK measures represent a helpful tool for the characterization and assessment of children with handwriting difficulties. The present study confirms that sex influences total BHK quality score, while school level influences handwriting speed.

## 1. Introduction

Children spend an average of 31 to 60% of their school time performing handwriting and other fine motor tasks [1]. Difficulties in fine motor tasks represent a common complaint among children in the general school population, and 11–12% of female students and 21–32% of male students are estimated to have handwriting difficulties, with a global prevalence of 10–34% of school-aged children [2,3]. Handwriting is indicated as an important school readiness skill and a predictor of the academic success [4,5,6], and it can be included among the core symptoms of the developmental coordination disorder (DCD) [7,8], with potential consequences in the academic progress, emotional well-being, and social functioning of the individual [9,10,11]. In fact, according to the DSM 5 [8], the DCD includes among its diagnostic criteria the delayed acquisition of motor milestones, with clumsiness, slowness, and inaccuracy in the performance of motor skills, both on the gross motor side and on the fine motor side. In addition, this motor skills deficit starts in the developmental age, and it significantly interferes with daily living activities. However, the DSM 5 does not discriminate between different subtypes of DCD, although individuals could be predominantly impaired in the gross motor skills as well as in fine motor skills, such as handwriting, which seems to have a predictive validity with respect to the diagnosis of DCD itself [12,13,14]. Moreover, handwriting difficulties affect daily living activities [15] and persist noticeably into adulthood [16], being frequently considered the reason for referral to occupational therapy [1,17] and task-based training [18]. Therefore, handwriting competence is commonly affected in these children, with frequent repercussions on the legibility and speed of written texts. Furthermore, handwriting has proven to be remarkably discriminative in children at risk of DCD at the Movement Assessment Battery for Children-2 (MABC-2) [19], which was validated in children aged 4–12 years and is currently considered the gold standard assessment tool used to identify impairments or delays in motor development. In particular, the MABC-2 includes 24 subtests and three domains (manual dexterity, balls skills, and balance). Results are usually interpreted as percentiles, with ≤5 percentile pinpointing a definite motor impairment, whereas ≤15 percentile suggests a borderline motor impairment. Another assessment instrument employed in clinical and research settings is the Developmental Test of Visual–Motor Integration (VMI) [20], which provides information about the visual and motor abilities of the subject, being considered a useful screening tool for DCD. In particular, the VMI contains three subtests oriented to explore different aspects of the symptom: visual motor integration, visual perception, and motor coordination. Results are interpreted as percentiles, with ≤5 percentile suggesting a clinical impairment which is worthy of a deeper evaluation. However, these tests are not devoted to handwriting and MABC-2 explores primarily gross motor skills; thus, other motor competences such as handwriting should be assessed separately through different standardized and psychometrically sound measures [21]. 

In order to assess handwriting skill in clinical practice and in experimental settings, the Concise Assessment Scale for Children’s Handwriting (BHK) is often used in a few countries [3,22,23,24,25,26]. It represents an analytic scale that provides a quick evaluation of handwriting quality and speed through a copying text. It can be used as a screening tool, as well as a diagnostic test. Instructions and normative values are derived from the Dutch original version [22]; references for normative values on quality of handwriting are available for children in grade two (age 7–8 years) and three (8–9 years), while references on speed are available for all grades [23]. The Dutch norms for writing speed derived from handwriting samples collected from 895 school children in first to sixth grade (ages roughly ranging between 78 and 150 months) [22]. The psychometric properties of the BHK were investigated by extensive research, and the scale was also adapted and validated for the French school population [25]. Other diagnostic tools are available in Italian for the assessment of handwriting skills, but they usually do not assess handwriting quality and investigate only writing speed. For example, the BVSCO explores writing speed of disyllables, single words, and numbers [27]. Furthermore, the BHK is preferred to the BVSCO due to its good psychometric properties, which were widely investigated in the literature, along with differences between boys and girls in handwriting quality and the deterioration of the form aspects of writing along the different grades of the primary school [28]. Moreover, in contrast to other tools, the BHK explores both handwriting quality and speed, and it is used in different languages and teaching methods, and, therefore, is able to be used extensively in clinical and research settings. A short version is the Systematic Screening for Handwriting Difficulties Test (SOS) [29], for which, however, sensitivity and specificity should also be investigated.

French students have l’ecriture cursive in their curriculum; the handwriting teaching method in this country involves the use of special papers with grids to ensure students keep their writing straight and at the right height and size. The Dutch handwriting teaching method plans the explicit sequencing of each letter movement, with the use of visual prompts such as the traffic light colors. The letter tracing is guided from the starting point (the green light) until the stop point (red light). Conversely, there are no clear guidelines regarding the timing and method for teaching handwriting in primary schools in Italy, the calligrafia is no longer in the curriculum of Italian students and teachers are free to choose different techniques [30]; global methods (without explicitly sequencing the movement of the letters) are widespread [31].

Considering the differences that are likely to occur between the Dutch, the French, and the Italian teaching methods in handwriting, the aim of the present study was to validate the Italian adaptation of the BHK [26] in a representative primary-school population in order to gather handwriting parameters values and provide Italian norms, comparing them with Dutch and French ones. The BHK demonstrated to be suitable for describing changes in the handwriting characteristics during the intervention monitoring [32], distinguishing between skilled writers and poor hand writers, but it has a broad borderline range. Previous authors labeled scores between 22 and 29 as at risk of poor handwriting [23] or ambiguous handwriting [32]. Secondly, the present study was aimed at exploring the distribution scores of the general population in order to suggest a cut-off score which allows through Z-scores for a better understanding of the level of impairment of the individuals in relation to the mean performances on the scale.

## 2. Materials and Methods

### 2.1. Study Design and Participants

The present cross-sectional study was conducted by physicians and other healthcare professionals, such as neuropsychomotor therapists of the developmental age. We randomly selected the 13.7% of the total children population of an area of Rome counting 4322 students attending the primary school; 16 public primary schools were identified out of a total of 22 in one of the largest municipalities of Rome (XX municipality), which includes both inner city and outskirts. Thus, this area was considered to be sufficiently representative of the socio-economic status (SES) of the population of Rome, since some studies showed that writing skills may be influenced by SES [33]. Children attending the second to fifth grade classes of primary school, with an age range between 7 and 11 years old, were included in the study whether they met the following inclusion criteria: (a) at least two years of schooling in Italy; (b) ability to write in cursive; and (c) ability of producing a text at least five lines long. We excluded from the sample students attending the first-grade class of primary school with the aim of avoiding the effect of factors related to practice and to the early stages of the learning of the written language. Seven to eight classes for each grade level were randomly selected and entirely assessed. The number of the selected classes was determined in proportion to the number of classes present in the districts and in the respective schools to which they belonged, and the classes were selected according to a simple random design.

### 2.2. Informed Consent and Ethical Approval 

All the followed procedures were in accordance with the ethical standards of the responsible committee on human experimentation, as well as with the Helsinki Declaration (1975, revised in 2008). The study was approved with the Prot. Number P-434-13 (30 October 2013) of the Pediatrics and Child Neuropsychiatric Department of Rome. The parents of all children enrolled in the study were informed about the research project by researchers and teachers through organized meetings at school and explanatory leaflets, and they provided their written informed consent.

### 2.3. Measures

Children’s task was to copy a text using cursive handwriting, within a time limit of 5 min, at the rate at which they usually wrote. Black roller pens were distributed in order to control the possible writing instrument bias. We excluded texts written in block letters and those less than five lines long. These handwriting samples were collected in May, at the end of the school year. Handwriting was tested with the Italian version of the Concise Assessment Scale for Children’s Handwriting [26]. The Italian copying text adaptation maintains the original design of the Dutch version [22]. It is in fact designed with a structure of increasing complexity: the first 5 lines contain simple monosyllabic words that children met in first grade; then, the text becomes more challenging, and the size of the letters decreases. The entire adaptation procedure was approved by the developer (Dr. Hamstra-Bletz, personal communication, January 2008). Back translation was not considered necessary due to the low effect of the meaning of words in the copy task. The handwriting task is to copy a text in 5 min on an unprinted A4 sheet. Four neuropsychomotor therapists of the developmental age trained in the scale administration procedures performed a copy test in group sessions. The same operators, after a few meetings aimed at homogenizing the evaluation criteria and establishing a consensus around each parameter of the scale, proceeded to evaluate the individual copy tests.

The BHK quality is evaluated through 13 parameters scoring from 0 to 5 (total score 0–65) according to legibility. The 13 parameters are: (1) letter size, (2) left-hand margin, (3) word alignment, (4) word spacing, (5) acute turns in joins or letters, (6) irregularities in joins, (7) collision of letters, (8) inconsistent letter size, (9) incorrect relative height of letters, (10) odd letters, (11) ambiguous letter forms, (12) correction of the letters, and (13) unsteady writing trace.

Three categories of writers were defined in the original BHK guidelines: children with a total score of 29 or higher are classified as non-proficient writers, those with a total score between 22 and 28 as at-risk writers, and children with a total score of 21 or less are classified as proficient writers [23].

On the other hand, BHK speed is calculated by counting the number of letters written in 5 min, excluding punctuation marks and including letters that have been erased. Four child therapists skilled in motor development assessments were specifically trained on BHK procedures in order to homogenize evaluation criteria on writing samples and ameliorate the agreement among them.

### 2.4. Statistical Analysis 

Data were analyzed with SAS v9.1 Statistical Package. Descriptive statistics were used to describe the sample population according to school level and gender. Internal consistency and test–retest reliability were calculated. Uniformity of circular (“directional”) data was assessed with Wilks’ λ (lambda) and with Rao’s ρ (rho) tests. Furthermore, MANOVA test for general variance and ANOVA were performed. Significant difference was assessed for *p* < 0.05.

## 3. Results

### 3.1. Sample

Overall, 594 children from 31 classes were enrolled in the study. Thirty-two participants (5.3%) used block letters or produced a sample shorter than five lines, and thus were excluded from the study. A total of 562 written samples were therefore analyzed. Children had a mean age of 9.3 years (SD 1.0; range 7.3–11.1); eight children (1.4%) had special needs and required a supporting teacher. Distribution of the sample is reported in Table 1. The Shapiro–Wilk Test was performed to assess the normal distribution of BHK quality scores, with a value of 0.979, compatible with a Gaussian distribution.

Internal consistency was calculated over the whole sample of 562 children, whereas test–retest reliability among the four evaluators was calculated considering only 144 writing pieces randomly selected and independently evaluated by each therapist.

At first, the reliability of the technicians’ scores was tested. Therefore, Kendall’s correlations for all pairs of technicians were performed for both parameters (BHK quality and BHK speed scores), and the results showed a sufficient to high concordance among technicians. Inter-rater reliability obtained through Kendall test was between 0.82 and 0.93 for writing speed, and between 0.42 and 0.63 for writing quality.

Moreover, Wilks’ λ and Rao’s ρ tests were performed for the uniformity of circular data (Table 2) and a MANOVA (multivariate analysis of variance) was applied, treating level and gender as independent variables and BHK quality and BHK speed as dependent variables.

P levels were highly significant (*p* < 0.00014); thus, the reliability of data was accepted and MANOVA was performed for both genders for each one of the two parameters, i.e., number of graphemes and BHK score.

### 3.2. Handwriting Quality (BHK Quality)

We obtained that 21.6% of the sample was classified as borderline/at-risk writer. The 10% (weighted value) of the sample is over the cut-off score of 29; these children (n = 56) are classified as non-proficient writers.

Table 3 describes the results of MANOVA performed on handwriting quality for genders and grades.

The effect of the variable gender on the handwriting quality was checked.

This analysis evidenced consistent differences for genders F (3.553) = 17.796 *p* = 0.000029. No consistent differences were found for grades, and neither was consistent interaction was observed.

BHK quality mean cumulative values were 20.3 with a standard deviation of 6.1 for boys, whereas they were 18.1 with a standard deviation of 6.7 for girls (Table 4). 

Scores which stood one SD and a half above the mean (SD × 1.5) were considered indicative of a poor handwriting quality, and thus used as cut-off values [30]. Therefore, the BHK quality cut-off values were 29.45 for boys and 28.15 for girls.

### 3.3. Copying Speed (BHK Speed)

Results of MANOVA performed on handwriting speed for genders and grades are shown in Table 5. The analysis indicated significant differences for school level: F (3.553) = 150.15; *p* < 0.00001.

No consistent differences were found for gender; no significant interaction between school level and gender was evidenced as well. These analyses showed similar results to the French study [34], where significant differences were obtained for school level as well, and French girls showed a higher copying speed than boys. Likewise, the Dutch studies on the argument [31,35] reported an increase in handwriting speed as the grades progressed (Table 6).

Then, we applied the Duncan test to evaluate class differences for the number of graphemes. According to the Duncan test, all grades were consistently different from other grades, whereas no consistent differences were found between genders.

The mean number of graphemes cumulated for boys and girls, for each grade, was 137.4 (SD 29.5) for second grade; 185.2 (SD 46.2) for third grade; 242.5 (SD 63.6) for fourth grade; and 276.9 (SD 80.3) for fifth grade. These values may be considered as indicative for class level in a normal control population. Scores below one standard deviation and a half for the grade should be considered indicative of a slow handwriting performance [29].

## 4. Discussion

Inter-observer reliability can influence the BHK total score (BHK quality), and therefore can influence the final judgment on the global legibility of a piece of handwriting; therefore, performing preliminary consensus sessions on ambiguous voices of the BHK scale can ameliorate the inter-rater reliability from moderate to good. In a previous study, inter-rater reliability varied between r = 0.71 and r = 0.94, depending on the grade level and number of evaluators [22,23]. For writing speed, the agreement is excellent and does not require consensus sessions. The included population followed a normal distribution for the BHK quality scores, as it was shown for the 837 children studied for the French validation of the scale [25]. The variables of sex and school level both influenced global BHK scores (quality and speed). In particular, sex, but not school level, influenced the total quality score; conversely, school level, but not sex, influenced the copying speed. This means that different normative values should be considered for sex to be used in the correction of a piece of handwriting at school age. Handwriting quality was higher in girls than in boys. This is consistent with other studies on the subject, on French and Dutch samples [25,28,36].

The results confirmed that the BHK total quality score is a stable parameter along the school years and does not vary significantly with regard to the years spent in the handwriting exercise, from second to fifth grade. Other authors have observed that gender affects the quality of writing [35], and that stability of handwriting quality occurs only in those children whose scores fall within the normal range. In fact, children with proper writing reached their final quality level at the end of first grade, while children with dysfunctional writing can improve significantly until second grade [23]. The French validation of the tool [25] found that school level and age had a significant effect on the BHK quality score; this can be explained by the fact that their sample included also 6- and 7-year-old children (at the beginning of primary school). We excluded from the sampling these age groups because children with typical development showed a rapid quality improvement in handwriting during the first grade (age 6–7 years old), with a stability reached by second grader (age 7–8 years old and over) [23], when handwriting had become more automatic and organized. Differences in the BHK quality scores between our data and the French sample (boys M = 15.2, SD = 6.6; girls M = 12.5, SD = 4.5) are probably explained by differences in handwriting teaching methods. The mean and SD values we obtained for BHK quality in boys and girls allow us to suggest 29 as the cut-off value for both genders; therefore, children obtaining a total score of 29 or higher are classified as non-proficient writers.

The handwriting speed is influenced by grade, and a significant difference for each of the grades from the second to the fifth was found. These data are to be considered in the norm referenced correction tables for speed, when child performance is compared (mean and SD). This finding is consistent with the previous data on the argument [25,28,36]. Speed values reported in this study in the BHK Italian sample [26], French sample [25], and Dutch sample [28] are compared with the writing speed of BVSCO [27] in subtests “le”, “uno”, and “numeri” (Figure 1).

These studies described a continuous and somehow linear increase in the writing speed through the second to the fifth grade. The comparison with second- and third-grade Dutch children [23] reveals a similar speed for the second and third graders (with a higher SD in the Dutch sample). Other instruments that measure the writing speed that are in use in Italy, such as the BVSCO, use a self-dictation of disyllables, single words, and numbers. Higher speeds shown in this test are probably in relation to a simpler task. 

The present study confirms that sex influences total BHK quality score (i.e., legibility) and that school level within primary school influences handwriting speed (i.e., fluency). However, this study presents some limitations. A first limitation is the high inter-rater variability which was found in the final BHK quality judgment, despite the use of specifically trained evaluators on the BHK scoring procedures; therefore, we underline the importance of maximizing the inter-rater consensus regarding BHK quality judgment, through a training among evaluators on ambiguous voices. In any case, a learning period is necessary to master the scoring system involved in final quality judgment. A second limitation of this study is the fact that we did not collect the exact socio-economical status of the students along the sampling, in absence of dedicated anagraphic cards. Consequently, the influence of this variable on legibility and speed was not investigated.

## 5. Conclusions

The use of both the BHK measures represents a helpful tool in the clinical practice for the characterization and the assessment of children with handwriting difficulties [31]. Thus, the present test could be useful in a developmental evaluation in association with other tools which investigate global and fine motor skills, such as the Movement Assessment Battery for Children (Movement-ABC) [19], or in place of other tests such as the Developmental Test of Visual–Motor Integration (VMI) [20], which has proved to be inadequate as a screening tool for children’s writing problems [23]. Therefore, the BHK is a reliable instrument in handwriting evaluation, and it should be considered as part of a larger neuropsychological investigation, as it is recommended for the DCD diagnostic process [37]. Nevertheless, the gold standard of any assessment is in the sum of all data resulting from the neuropsychological testing and the clinical evaluation of the child.

## Figures and Tables

**Figure 1 children-10-00223-f001:**
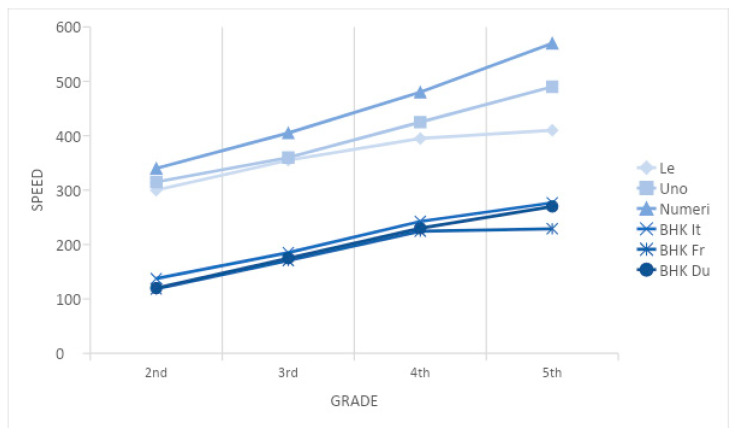
Comparison among the speed in three different versions of BHK (Italian, French, and Dutch) and the three subtests of the BVSCO (“le”, “uno”, and “numeri”) for school level.

**Table 1 children-10-00223-t001:** Number of children of the sample divided for grade level and gender.

Grade	Gender		
	Male N (%)	Female N (%)	Total N
2nd	73 (55)	59 (45)	132
3rd	57 (48)	62 (52)	119
4th	81 (52)	74 (48)	155
5th	72 (47)	84 (53)	156
Total	283 (50)	279 (50)	562

**Table 2 children-10-00223-t002:** General evaluation of the sample.

EFFECT	Wilks’ λ	Rao’s ρ	dF 1	dF 2	*p*-Level
Class	0.545400	65.14954	6	1104	0.000000 *
Gender	0.968243	9.05249	2	552	0.000135 *
1, 2	0.995753	0.39194	6	1104	0.884502

Note: 1 is for class; 2 is for gender; Wilks’ λ (lambda); Rao’s ρ (rho). * indicates significant differences.

**Table 3 children-10-00223-t003:** Handwriting quality (BHK quality): General MANOVA.

	dF	MS	dF	MS		
**EFFECT**	**EFFECT**	**EFFECT**	**ERROR**	**ERROR**	**F**	***p*-Level**
1	3	75.97	553	41.76	1.819	0.1425
2	1	743.16	553	41.76	17.796	0.000029 *
1, 2	3	16.11	553	41.76	0.386	0.7632

Note: 1 is for grade; 2 is for gender. * indicates significant differences.

**Table 4 children-10-00223-t004:** BHK quality mean values by gender.

	Total	Males	Females
Mean	19.3	20.3	18.1
SD	6.5	6.1	6.7

Note: SD: Standard Deviation.

**Table 5 children-10-00223-t005:** Copying speed (BHK speed): General MANOVA.

	dF	MS	dF	MS		
**EFFECT**	**EFFECT**	**EFFECT**	**ERROR**	**ERROR**	**F**	***p*-Level**
1	3	531,550	553	3,540,189	150.14	0.000000 *
2	1	1822.4	553	3,540,189	0.5148	0.4733
1, 2	3	1448.9	533	3,540,189	0.4093	0.7464

Note: dF = degrees of Freedom, MS =Mean Squares; 1 is for grade; 2 is for gender. * indicates significant differences.

**Table 6 children-10-00223-t006:** Comparison among the speed in three different versions of the BHK (Italian, French, and Dutch) for school level.

School Grade	BHK It	BHK Fr	BHK Du
2nd	137.4	118.3	120
3rd	185.2	170.5	175
4th	242.5	224.5	230
5th	276.9	229.1	270

Note: all values are considered for the end of the school year.

## Data Availability

The data presented in this study are available on request from the corresponding author.

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
