# Peer review of "Validation of the Concise Assessment Scale for Children’s Handwriting (BHK) in an Italian Population"

_children, 2023, doi:10.3390/children10020223_

Round 1

Reviewer 1 Report

Thank you for the opportunity to review this manuscript. In attachment you can find my recommendations.

Author Response

Thank you for your comments. We revised the abstract of the study and added information about the significance of the results. Furthermore, we expanded the introduction section and highlighted the relevant aspects of the tool and its potential in clinical and research settings (please see lines 38-43 and 60-63). In addition, we explained the study design in the materials & methods section and added the ethical approval details (lines 99-100). Moreover, we explained the inclusion criteria of the recruitment (lines 88-91). We also included in the results Table 5, which compares speed in the Italian, French, and Dutch sample (lines 213). Finally, we added the conclusions section, as kindly requested.

Reviewer 2 Report

This manuscript describes the adaptation of the BHK test on hand-writing quality to the Italian population. It describes the effect of gender on overall correctness scores, and of grade (1-5th) on handwriting speed.

I have a couple of minor questions or suggestions that I would like the authors to consider:

1. In the introduction, please explain more clearly how the quality of handwriting can be linked to other fine motor skills and can thus be used as a reliable indicator.

Moreover, can it be said that "it causes a significant delay in learning achievement of motor skills", or it is rather a symptom or predictor (as suggested by [5]? Otherwise, references 4-6 do not support the claim in lines 32-33.

2. Please describe the Italian handwriting teaching method, and why this can be causing differences with the French and Dutch population.

3. It is said that the sample is representative, but it is not adequately justified (l. 59). What is the population of reference? Children living in Italy or in Rome? the distribution of socio-economic status in the population and the sample? Is this variable even related to handwriting quality, and is it the only one?

4. Please consider including a revision of other existing tools, and explain why this one is better, and is the one that should be included in more extensive diagnosis procedures.

Author Response

Thank you for your comments. We revised the introduction section and explained more thoroughly how handwriting could be useful to assess fine motor skills and therefore be a reliable indicator for DCD (please see lines 38-43). Moreover, we rephrased the sentence you commented (lines 34-35) in order to clear the contradiction. In addition, we described the Italian handwriting teaching method (see lines 63-72), as kindly requested. Furthermore, we clarified in the Materials & Methods section which is the population of reference, and explained why the socio-economic status may be relevant (lines 86-88). Finally, we reported other similar tools employed in clinical settings, explaining why the BHK may be more accurate in this specific area (lines 57-63).

Round 2

Reviewer 1 Report

The authors improved the manuscript according with the recommendations.